# The Safety and Efficacy of New DIVA Inactivated Vaccines Against Lumpy Skin Disease in Calves

**DOI:** 10.3390/vaccines12121302

**Published:** 2024-11-21

**Authors:** Gaetano Federico Ronchi, Mariangela Iorio, Anna Serroni, Marco Caporale, Lilia Testa, Cristiano Palucci, Daniela Antonucci, Sara Capista, Sara Traini, Chiara Pinoni, Ivano Di Matteo, Caterina Laguardia, Gisella Armillotta, Francesca Profeta, Fabrizia Valleriani, Elisabetta Di Felice, Giovanni Di Teodoro, Flavio Sacchini, Mirella Luciani, Chiara Di Pancrazio, Michele Podaliri Vulpiani, Emanuela Rossi, Romolo Salini, Daniela Morelli, Nicola Ferri, Maria Teresa Mercante, Mauro Di Ventura

**Affiliations:** 1Istituto Zooprofilattico Sperimentale dell’Abruzzo e del Molise “G. Caporale”, 64100 Teramo, Italy; f.ronchi@izs.it (G.F.R.); a.serroni@izs.it (A.S.); m.caporale@izs.it (M.C.); l.testa@izs.it (L.T.); c.palucci@izs.it (C.P.); daniela.antonucci1@alice.it (D.A.); s.capista@izs.it (S.C.); s.traini@izs.it (S.T.); c.pinoni@izs.it (C.P.); ivano.dimat@gmail.com (I.D.M.); c.laguardia@izs.it (C.L.); g.armillotta@izs.it (G.A.); f.profeta@izs.it (F.P.); f.valleriani@izs.it (F.V.); elisabetta.difelice@asl2abruzzo.it (E.D.F.); g.diteodoro@izs.it (G.D.T.); f.sacchini@izs.it (F.S.); m.luciani@izs.it (M.L.); c.dipancrazio@izs.it (C.D.P.); m.podaliri@izs.it (M.P.V.); e.rossi@izs.it (E.R.); r.salini@izs.it (R.S.); d.morelli@izs.it (D.M.); nicolaferri1955@gmail.com (N.F.); t.mercante@izs.it (M.T.M.); m.diventura@izs.it (M.D.V.); 2Department of Veterinary Medicine, University of Teramo, 64100 Teramo, Italy

**Keywords:** lumpy skin disease virus, inactivated vaccine, DIVA

## Abstract

**Background:** Lumpy skin disease virus (*Poxviridae* family—*Capripoxvirus* genus) is the aetiological agent of LSD, a disease primarily transmitted by hematophagous biting, affecting principally cattle. Currently, only live attenuated vaccines are commercially available, but their use is limited to endemic areas. There is a need for safer vaccines, especially in LSD-free countries. This research aims to develop and test a safe and efficacious inactivated vaccine. Moreover, in this study, we used keyhole limpet hemocyanin (KLH) as a positive marker to distinguish infected from vaccinated animals (DIVA). **Methods:** Lumpy skin disease virus was propagated on primary lamb testis cells and Madin–Darby bovine kidney cells (PLT and MDBK, respectively), and four inactivated vaccines were produced. The vaccines differed from each other with the addition or not of KLH and in cells used for virus propagation. To evaluate the safety and immunogenicity, the vaccines and two placebos were administered to six groups comprising six male calves each, and antibody response was investigated using both an enzyme-linked immunosorbent assay (ELISA) and a serum neutralization (SN) test. In addition, the LSD/γ-interferon test and KLH (IgM-IgG) ELISA were performed on the collected samples. Furthermore, the use of KLH allowed us to distinguish vaccinated animals in the ELISA results, without any interference on the strength of the immune response against the LSDV. Finally, the efficacy of one of four vaccines was investigated through a challenge, in which one group of vaccinated animals and one animal control group were infected with a live field strain of LSDV. **Results:** Four out of the six control animals showed severe clinical signs suggestive of LSD, and, therefore, were euthanized for overcoming the predetermined limit of clinical score. By contrast, the vaccinated animals showed only mild symptoms, suggesting a reduction in severe disease notwithstanding the incapability of the vaccine in reducing the virus shedding. **Conclusion:** The vaccines produced were safe and able to elicit both a humoral and a cellular immune response, characteristics that, together with the demonstrated efficacy, make our vaccine a good candidate for countering the LSD spread in disease-free countries, thus also facilitating disease containment throughout the application of a DIVA strategy.

## 1. Introduction

Lumpy skin disease (LSD) is a viral disease that naturally affects cattle, especially the young and cows in the peak of lactation [1,2]. The aetiological agent of LSD, called lumpy skin disease virus, belongs to the *Poxviridae* family and *Capripoxvirus* genus, and it is primarily transmitted through hematophagous biting midges [3], which act as mechanical vectors rather than biological ones [4,5]. Lumpy skin disease is usually characterized by a low mortality rate (up to 10%), and different clinical signs have been reported. The signs may vary from mild, subclinical forms to more severe symptoms, such as ocular and nasal discharge, high fever, and multiple nodular skin lesions with subsequent poor skin quality. In addition, abortion, temporary or permanent infertility, and a strong failure in milk production [6] have also been recorded. These latter symptoms are the most important causes of the huge economic losses related to this viral disease in the cattle industry [2]. Lumpy skin disease appeared for the first time in 1929 in Zambia and rapidly became endemic in most African countries [1,6,7]. Since 2012, many LSD outbreaks have been reported in the Middle East, until the first LSD incursion occurred in Southeastern Europe (Greece, Bulgaria, Serbia, Albania, and Montenegro) during 2015–2016 [7,8]. To contain LSD in Europe, authorities have carried out different countermeasures: culling all infected animals, prohibiting animal movement, and requiring vaccination. Vaccination is the most effective strategy not only to limit LSD spread but also to eradicate the disease [2,7,9]. Currently, the commercially available vaccines against LSD are live attenuated ones. The use of these vaccines is suggested and limited to endemic countries. Their limited use is related to adverse reactions [10,11,12] and the risk of vaccine virus shedding from skin lesions and milk [10,13]. Moreover, the potential risk of viral reversion and recombination between the wild field strain and the attenuated virus of the vaccine contributes to limiting the use of these types of vaccines [14]. Thus, there is a need for advancement in the production of safer vaccines, like inactivated ones, to be specifically used in LSD-free countries to counter the spread of the disease. In the last decade, different efforts have been made in order to produce safe inactivated vaccines against LSDV [15,16,17,18,19,20], but it should be highlighted that none of them enables discrimination between vaccinated from naturally infected animals, which is the principle of the DIVA strategy [21,22,23,24]. The application of this strategy in the vaccinology field represents an innovative helpful instrument to contrast, control, and eradicate a disease, and it is carried out by means of two tools: (i) a vaccine that contains a positive or negative marker, or both, and (ii) a validated diagnostic test. In the past, keyhole limpet hemocyanin (KLH), from a giant gastropod *Megatura crenulata*, was used as a positive marker in a DIVA bacterial vaccine [25]. Together with the marker properties, KLH has immunostimulatory properties, activating both humoral and cellular immune responses [25,26]**,** and has therefore been used as a protein carrier and adjuvant in vaccines [26,27,28]. The aim of our research was to develop a safe, immunogenic, and efficacious inactivated vaccine useful for identifying vaccinated animals by KLH employed as an exogenous protein marker, providing a tool for countering the spread of LSD in both endemic and non-endemic countries.

## 2. Materials and Methods

### 2.1. Ethics Statement

The animal experiments were approved by the Italian Ministry of Health (Direzione Generale della Sanità Animale e dei Farmaci Veterinari, Ufficio 6; Authorization n. 807/2020-PR of 10 August 2020) and were conducted at the animal facility (944F0) of the Istituto Zooprofilattico Sperimentale dell’Abruzzo e del Molise “G. Caporale”, Teramo, Italy, where the farm owning the authorization for animal experimentation (Authorization n. 14/2021-UT art. 20 D.lgs 26/2014) is located.

### 2.2. Animals

The experiment was carried out on 36 healthy male Holstein–Friesian calves, purchased from different farms in the Abruzzo Region (Italy). *Bos taurus* was chosen since it is the most sensitive species to LSD [29]**,** and no sex differences are reported on susceptibility to disease. As recommended by Regulation (EU) 2019/6 [30], the experiment was performed on young animals (4–8 months old), which represent the age category most susceptible to LSD [20]. The animals were kept in free-stall conventional housing with straw as bedding, feed, and water provided ad libitum; species-specific environmental enrichment was available (contact with conspecific and human, cow brush, salt blocks, and blue balls). They were divided into 6 groups (A–F) of 6 animals each, using computer-generated numbers (Microsoft Excel 2013, 15.0.5475.1000, Windows 11) (Table 1).

The challenge procedure was carried out on 12 animals (6 vaccinated animals and 6 animals in the placebo group), inside the BSL3 facility, where the values of temperature (16–28 °C), pressure (−10 Pa), and humidity percentage (30–70%) were systematically assured. A system of internal video cameras with remote connection by phone allowed for the 24/24 h observation of hospitalized animals without the need to enter the housing premises. Traps with attractive lamps were installed in each box to verify the absence of flying insects. Clinical score sheets were used in daily animal management to assess the severity and progression of the induced disease, ensuring improvement in animal welfare. The observations were structured on the following four high-level categories: appearance (body condition, coat and skin condition, and discharge from eyes and mouth), body functions (respiration, food/water intake, and temperature), environment (enclosure environment, including any litter, material, and enrichment items), and behaviors (social interaction; undesirable behaviors; posture and mobility; tremors; seizures/convulsions/spasms; and vocalization, spontaneous or invoked). Moreover, through the implementation of the humane endpoint, set at the value of 13, animal pain and distress were relieved while still meeting experimental objectives.

### 2.3. Virus Origin and Isolation

The field viral strain used for vaccine production and the challenge test was isolated from a symptomatic calfskin nodule, as described by Babiuk et al. [31]. The sample was collected during an outbreak that occurred in Albania in 2017 and was kindly provided by Dr. Ledi Pite and Dr. Liljana Cara from the Food Safety and Veterinary Institute (FSVI) in Tirana. Briefly, the biopsy sample was weighed, ground in a mortar, and suspended 1:10 (*w*/*v*) in phosphate-buffered saline (PBS) pH 7.2, supplemented with 10,000 IU/mL penicillin (Sigma, Burlington, MA, USA), 5000 IU/mL nystatin, 10 g/L streptomycin sulfate salt (Sigma, Cat# S9137), and 125 mg/L gentamycin sulfate (Sigma, Cat# G1397). The suspension was frozen at −80 °C, thawed three times, and centrifuged at 3000× *g* for 30 min at 4 °C, and the supernatant was used for virus isolation. The virus stock was obtained after five consecutive passages on OA3.Ts cells (ATCC Cat# CRL-6546, RRID: CVCL_3764), a continuous cell line susceptible to LSD virus infection. Cells were incubated at 37 °C, 5% CO_2_, and observed daily using an inverted microscope (20×–40× Leica DFC425 C, Leica Microsystem Ltd., Wetzlar, Germany) to evaluate the presence of virus-specific cytopathic effects (CPEs). When the CPE evaluation was complete, cells were frozen at −80 °C, thawed three times, and centrifuged at 3000× *g* for 30 min at 4 °C. The supernatant was then collected, distributed into aliquots, and stored at −80 °C. One aliquot of the supernatant was checked for sterility for bacteria, fungi, and mycoplasma as prescribed in the European Pharmacopoeia (Ph. Eur. 11.0, 20601 (04/2011); anonymous); another aliquot was used to determine the viral titer to identify the virus using real-time PCR and evaluate its purity, as prescribed in the European Pharmacopoeia [32]. Finally, a third aliquot of the virus was used to initiate vaccine production.

### 2.4. Inactivated Vaccine and Placebo Production

Four different inactivated vaccines against LSD and two placebos were produced (Table 1). Vaccines were prepared by amplifying the field viral strain in two different cell substrates: PLT and MDBK cells (ATCC Cat# CRL-6071, RRID: CVCL_0421). These two cell types are commonly used for LSD vaccine preparations, because of the high viral yield that they enable to obtain [33]. We carried out experiments involving vaccine stock production on both cell types in order to establish better cell substrates for viral propagation and vaccine production using the virus strain selected. The PLT cells were isolated and amplified in our lab following the method described by Mahmood [34]. In particular, the viral suspensions were obtained by infecting both MDBK and PLT cells with 0.1 MOI of a viral suspension produced starting from the freeze-dried working seed virus previously produced and checked. Briefly, the virus was suspended in a minimum essential medium (MEM, Gibco, Waltham, MA, USA Cat# 11095080) at a final concentration of 0.1 MOI and was adsorbed for 2 h on the monolayers of the two cell lines with monolayer at 80% of growth. After adsorption, a volume of 40 mL of MEM was added to each 175 cm^2^ flask. Then, the flasks were incubated at 37 °C at 5% CO_2_ and observed daily at the inverted microscope (20×–40×). The viral suspensions were harvested when the CPE was estimated to be equal to 90–100% and centrifuged at 3000× *g* for 30 min at 4 °C, and the supernatants obtained were stored at 4 °C until use. At the same time, the pellet obtained was suspended in a volume of PBS, pH 7.2, equal to 1/10 of the starting volume, and underwent the freeze–thaw process at −80 °C three times. Next, the viral suspension obtained from the pellet was centrifuged at 2000× *g* for 30 min at 4 °C and was combined with the previously harvested supernatants. The two viral suspensions were checked in terms of their titer; identity; purity; and sterility for bacteria, fungi, and mycoplasma, as prescribed by the European Pharmacopoeia [32]**,** and stored at 4 °C until use. The viral suspensions obtained from the two cell lines were subjected to concentration/purification using ultrafiltration cassettes with a nominal cut-off of 300 kDa (Merck Millipore, Burlington, MA, USA, Cat# P2C300V01) using the Cogent M1 TMP System instrument (Millipore). The virus was concentrated 10× and used for the production of four different inactivated vaccine formulations, as described by Luciani et al. [35]. Briefly, a solution of 100 mM binary ethylenimine (BEI) was prepared according to Bahnemann [36] and used at a final concentration of 5 mM in order to inactivate the viral suspensions, incubating them at 37 °C for 24 h. Then, 20 mM sodium thiosulphate (Sigma, Cat# 563188) was added, and the suspensions were stored at 4 °C until use. The inactivation control was performed inoculating OA3.Ts cells, grown in 75 cm^2^ flasks, with samples of inactivated suspensions. Once verified the inactivation, the viral suspensions, previously concentrated, were emulsified with 10% (*v*/*v*) Montanide Gel 01 PR (gently provided from Seppic, Paris, France) and supplemented with 0.03 mg/mL saponin (Sigma, Cat# SAE0073). Finally, 20 µL/mL of KLH (Sigma, Cat# H8283) was added in two out of the four vaccines obtained and in one out of the two placebos in order to obtain a KLH concentration equal to 130 µg/mL. In conclusion, the vaccines differed from each other in the cell line used (PLT or MDBK) and the presence or absence of KLH, used as an exogenous marker protein. Two placebos containing PBS, Montanide Gel, saponin with or without KLH, were also produced (Table 1). The six formulations were verified for sterility for bacteria, fungi and mycoplasma before being used in the animal experiments.

### 2.5. Inoculation of Calves and Safety Study

The calves were allowed to acclimate to the new environment and feed for one month. To calculate the range of physiological temperature, rectal temperatures were taken daily for one week before inoculating the vaccine, and then the standard deviation of all values obtained was added and subtracted from the main value. A temperature of 40 °C was set as pyrexia. On day 0 of the experiment, the animals were moved one at a time to a containment cage; their rectal temperatures were measured; blood samples were collected, as described in the next section; and then the vaccine formulations were injected intramuscularly. Each group of animals was inoculated with one of the six formulations (Table 1). According to the recommendation of the European Pharmacopoeia [32], concerning the evaluation of the safety of veterinary vaccines, the animal rectal temperatures were taken 4 h later. The procedure described above was repeated on day 28, when the animals underwent a second vaccine inoculation. To assess the safety of the vaccine formulations, local reactions at the inoculation site and rectal temperatures of the animals were monitored for 14 consecutively days after each of the two doses administered.

### 2.6. Immunogenicity Studies

To determine the immunogenicity of the four vaccine formulations, the animals underwent weekly blood sampling from day 0 to day 63. Next, the blood sampling was performed monthly until day 385 only for the four animal groups not involved in the challenge test (groups A, B, D, and E). The immune response elicited against LSDV was analyzed by an ELISA, SN test, and in vitro interferon-γ test, as described below. The immunogenicity study for KLH was performed by analyzing the samples of the animal groups inoculated with the formulations containing KLH (groups A, C, and F) throughout an ad hoc ELISA. Blood sampling was performed from the jugular vein using a vacutainer holder and blood collection tubes, with and without anticoagulant. The blood samples without anticoagulant were incubated at 37 °C for 1 h, and then at 4 °C overnight. Next, the tubes were centrifuged at 1328× *g* for 10 min at 4 °C, the sera were collected, and the immunogenicity tests were performed. The blood samples contained in tubes with the anticoagulant (lithium heparin) were used to perform the interferon-γ test, as described below.

### 2.7. Serological Analyses

#### 2.7.1. LSDV ELISA and SN

The enzyme-linked immunosorbent assay was performed using ID Screen Capripox Double Antigen Multi-species ELISA (Innovative Diagnostics, Montpellier, France, Cat# CPVDA) according to the manufacturer’s instructions. Samples with an S/P % ratio value ≥ 30% were considered positive. To detect the neutralizing antibodies, the SN tests were carried out following the method described in the Terrestrial Animal Health Code of the World Organization for Animal Health [37]. For this purpose, serum samples were inactivated at 56 °C for 30 min and diluted serially 2-fold, from 1:5 to 1:640, in serum-free MEM in duplicate, using a 96-well microtiter plate. Each well contained 50 µL of serum dilutions, which were titrated against the LSDV Neethling strain (50 µL) with a fixed titer of 100 TCID_50_. The microtiter plates were incubated at 37 °C for 2 h and 5% CO_2_, and then the serum-virus suspensions were transferred onto a confluent OA3.Ts monolayer with a cell density of 10^5^ cell/mL and incubated again at 37 °C with 5% CO_2_. Plates were observed daily using an inverted microscope (20×–40× Leica DFC425 C, Leica Microsystem Ltd.) to detect the presence of virus-specific CPE. After 7 days, the plates were analyzed, and the neutralization titers were calculated as the highest dilution that completely inhibited CPE in over 50% of the wells. A serum with a titer of 1:10 or greater was considered positive.

#### 2.7.2. Interferon-γ Assay

The in vitro system is based on stimulation of the peripheral blood mononuclear cells (PBMCs) exposed to the antigen produced, as described below. The antigen, with a viral titer of 10^6.4^ TCID_50_/mL, was collected from PLT cells, concentrated with Amicon Ultra centrifugal filters 100 kDa MWCO (Merck Millipore, Cat# UFC9100), and suspended in PBS pH 7.2. Next, the concentrated virus was centrifuged on a 35% sucrose cushion at 35,000× *g* for 30 min at 4 °C. The pellet was suspended in PBS pH 7.2 and kept under magnetic stirring at 100 RPM for 3 h at 4 °C. Subsequently, the viral antigen was inactivated with 5 mM BEI, as previously described. Bovine IFN-γ ELISA (Mabtech, Stockholm, Sweden, Cat# 3119) was performed according to the manufacturers. Briefly, 3 aliquots of 1 mL of whole blood were stimulated in 24-well culture plates with 100 µL of viral antigen; 100 µL of mitogen (10 μg/mL of lectin concanavalin A Sigma), as positive control; and 100 µL of PBS as negative control at overnight (median 18 h). Next, the concentrations of IFN-γ (pg/mL) were calculated by interpolating the optical densities, detected at 450 nm wavelength, with the calibration standard curve, using a second-degree polynomial equation. All samples and standards were analyzed in duplicate.

#### 2.7.3. KLH Antibodies ELISA

The analysis of antibodies against KLH was performed using the commercial Bovine Anti-Keyhole Limpet Hemocyanin (KLH) IgM ELISA Kit and Bovine Anti-Keyhole Limpet Hemocyanin (KLH) IgG ELISA Kit (ALPHA DIAGNOSTIC INTERNATIONAL, San Antonio, TX, USA, Cat# 700–165-KBM and Cat# 700–160-KBG, respectively). A panel of sera from non-immunized animals (day 0) was tested at different dilutions (from undiluted to 1:1000 and from undiluted to 1:300 for IgG and IgM, respectively) to establish the serum sample dilution to use in ELISA. Once the optimal dilution was determined, all the samples were tested in both ELISAs, and the results were analyzed as reported in the manufacturer’s instructions.

#### 2.7.4. Efficacy Studies

Two animal groups (the vaccinated group C and the placebo group F, Table 1) underwent the challenge test, which was performed at day 63 by inoculating 7 mL of a live field virus strain (Albanian strain 7416/5), with a viral titer equal to 10^6.6^ TCID_50_/mL. The virus was administered through two routes, namely 0.5 mL of virus suspension was used for four intradermal injections in the neck region, and 5 mL was used for one intravenous injection through the jugular vein. Rectal temperatures, local reactions, and clinical examination results were recorded daily throughout the challenge experimental period. In the meantime, blood samples were collected every three days until the end of the experiment from the jugular vein, using a vacutainer holder and a blood collection tube with (lithium-heparin for interferon-γ and EDTA for real-time PCR) and without anticoagulant. The antibody responses against LSDV and KLH, and interferon-γ concentration during the challenge were investigated using the same methods described in the immunogenicity study. In addition, blood samples; swabs from the eyes, nares, and mouth; and skin biopsies, if nodules on the skin were present, were analyzed via PCR to determine viremia. The real-time PCR was performed to detect LSD viruses. Briefly, DNA from blood was directly extracted, and ocular, nasal, and oral swabs were first frozen at −80 °C and thawed 3 times. In the meantime, skin biopsies were homogenized in PBS pH 7.2, supplemented with antibiotics (10^6^ IU/L penicillin, 10 g/L streptomycin, 5 × 10^6^ IU/L nystatin, and 125 mg/L gentamicin; IZSAM, Teramo, Italy) using a Tissue Lyser II tissue homogenizer (QIAGEN, Hilden, Germany). DNA was extracted from blood, swabs, and homogenized skin biopsy samples using BioSprint 96 One-For-All Vet Kit (Indical Bioscience, Leipzig, Germany, Cat# SP947057) following the manufacturer’s instructions. Then, all the samples were tested using the pan-capripox real-time qPCR [38].

### 2.8. Statistical Analyses

A non-parametric Mann–Whitney test was applied to compare the results of each vaccinated group with the control groups. All tests were performed using R software v.4.0.5 (Core Team, Jaipure, RAJ, India).

## 3. Results

### 3.1. Viral Isolation and Inactivated Vaccines and Placebo Production

The field virus used for vaccine production, isolated from a biopsy in the OA3.Ts cells, had a titer of 10^5.3^ TCID_50_/mL. The titers of the virus used for vaccine production, before inactivation and 10× concentration, were 10^5.6^ TCID_50_/mL and 106.1 TCID50/mL in MDBK and PLT cells, respectively. The identity and purity tests produced favorable results and the inactivation controls demonstrated the absence of virus replication. In addition, all the six formulations listed in Table 1 proved to be sterile for bacteria, fungi, and mycoplasma.

### 3.2. Inoculation of Calves and Safety Study

The range of physiological temperatures was between 38.2 °C and 39.5 °C. Following vaccine inoculations, no local reactions were observed in any of the animal groups. Nevertheless, after the first inoculation, some febrile episodes were detected in the animals at different intervals post-vaccination (p.v.) (Figure 1A–C). In particular, two animals from the vaccinated groups B and C, one from the placebo group E, and three animals from the placebo group F experienced fever at 4 h p.v. Other fever spikes were detected during the experiment, but only two animals (groups B and C) had fever for two consecutive days.

The temperature rises detected 4 h after the first inoculation concerned two of the three formulations containing KLH, but the statistical analysis did not reveal any significant differences in the temperature values between the vaccinated and control groups (*p* > 0.05).

Some fever peaks were also detected after the second inoculation: two animals belonging to the placebo group E had fever at 4 h p.v., and other fever spikes were measured in five animals, with three belonging to the placebo group F, at day 5 and day 6 (Figure 1D–F). In the other days, the rectal temperatures of all animals remained under the value set as pyrexia (40.0 °C). The fever spikes observed after 4 h p.v. could be a consequence of the animal containment necessary for the inoculations.

### 3.3. Immunogenicity Studies

#### 3.3.1. LSDV ELISA

To evaluate the vaccine immunogenicity, an ELISA was performed on the cattle sera sampled, as previously described. After the first dose, only two animals in group D, vaccinated with MDBKv, were positive on day 28 (Figure 2B). All the vaccinated animals became positive a week after the administration of the second dose (day 35). The peak of S/P % values was reached at day 42 for all vaccinated groups (Figure 2A,B).

Although some animals became negative over time, the mean S/P % values remained above the cut-off for the entire duration of the immunogenicity study (up to day 385).

In particular, the S/P % values of three animals in group A, vaccinated with PLTv-KLH, ranged from negative to weakly positive starting from day 161. Likewise, only one animal in group B, vaccinated with PLTv, was negative starting from day 133 until the end of the study. The animals in group C, vaccinated with MDBKv-KLH, were all positive at day 63, when they underwent the challenge. Finally, all the animals in group D, vaccinated with MDBKv, maintained a positive immune response until day 385.

The statistical analysis showed significant differences in S/P % values between the four vaccinated animal groups and the control groups starting from day 7 for MDBKv, day 14 for PTLv-KLH and PTLv, and day 21 for MDBKv-KLH (*p* < 0.05). These differences persisted throughout the experimental period (day 385).

#### 3.3.2. LSDV SN

The immunogenicity of the four formulations was additionally investigated by analyzing animal sera with the SN test. All vaccinated animals were negative in SN until day 28 (Figure 3A,B). Next, seven days after the second inoculation (day 35), only two animals belonging to group C were negative, while all the other animals had positive titers ranging from 1:10 to 1:80. All the animals became positive at day 42, with titers ranging from 1:10 to 1:160. The highest values for all vaccine formulations in SN were reached at day 42, as also occurred in ELISA results.

For the entire period of immunological study, the mean value of log_10_ SN titer was >0 in all animal groups, except at day 161, in which all animals belonging to group A were negative. Equally, animals in the control groups were negative throughout the study. The statistical analysis (Mann–Whitney) showed significant differences (*p* < 0.05) between vaccine and placebo groups starting from day 35.

#### 3.3.3. Interferon-γ

The blood samples of vaccinated animals stimulated with the inactivated LSD antigen, as previously described, showed an increase in the interferon-γ concentration starting from day 7. By contrast, the interferon-γ concentration in the blood of control animals was equal to zero for the entire experimental period with the exception of some values that differed significantly from the mean, likely due to a technical issue (Figure 4A–D). The differences observed in interferon-γ concentration between the vaccinated and sham vaccinated animals were significant (*p* < 0.05) starting from day 28 and day 35 for PTLv-KLH against placebo and placebo–KLH, respectively (Figure 4A), from day 7 for PTLv against the placebo and day 14 against the placebo–KLH (Figure 4B), from day 28 for MDBKv-KLH against both placebos (Figure 4C) and from day 7 for MDBKv against the placebo and from day 14 against placebo–KLH (Figure 4D).

#### 3.3.4. KLH Antibodies

KLH was added in order to distinguish vaccinated animals. The O.D. values of IgM ELISA in animal groups A, C, and F, inoculated with KLH, followed the same trend (Figure 5A,B).

In particular, all the samples were negative at day _0_ and became positive seven days post-injection. Then, the number of positive samples decreased in each animal group (A, C, and F) until boost day (day 28). After the booster, all animals were positive at days 35, 42, and 49. Next, on day 63, three animals were found negative in each of the vaccinated groups (A–C).

No significant statistical differences (*p* > 0.05) in the O.D. values between the PTLv-KLH and placebo–KLH were found until challenge day (day 63)_._ In contrast, starting from day 42, the O.D. values of the animals inoculated with placebo–KLH were significantly higher (*p* < 0.05) than the values of animals vaccinated with MDBKv-KLH, until the day of the challenge (day 63). The animals in group A that did not undergo the challenge remained IgM-positive until day 161.

As expected, different results were obtained when analyzing the same samples with KLH IgG ELISA kit, (Figure 6A,B), which were negative at day 0 and became positive seven days post-injection, until day 14.

On days 21 and 28, only one animal in each vaccinated group (A–C) was found negative. After the booster, all animals were positive and remained as such until the day of the challenge (day 63). All the samples reached a high titer at day 35, which was maintained for the entire study duration (day 308). The O.D. values in the control group F were higher than in the vaccinated group (A and C), with statistically significant differences on days 14, 28, 42, and 63 (*p* < 0.05).

The results of the unchallenged group A showed KLH IgG positivity throughout the experiment; by contrast, the IgM titer was positive until day 68 and decreased to a negative value by the end of the study.

### 3.4. Challenge Study

#### Efficacy Studies

Since no significant differences were observed in the antibody response of the vaccinated groups, the challenge was performed in animals of only two groups (C and F) vaccinated with MDBKv-KLH and placebo–KLH (Table 1) in order to reduce the number of animals and limit the pain. The vaccinated group was selected considering that, for a hypothetical industrial vaccine production, the use of a continuous cell line could be preferable with respect to a primary cell line. In addition, the characteristics determined by the KLH presence could be an added value for a commercial vaccine. The control group was selected as a consequence of the vaccinated one. The challenge test was performed using a field viral strain as previously described. After infection, five animals in group F showed fever within day 6, while only one animal in group C had fever at day 5, for five consecutive days (Figure 7).

The clinical score values were measured daily and showed a constant rise in symptoms in all animals of the control group F, which may be attributed to lumpy skin disease, until day 9 (Figure 8), starting from the first day after infection. On day 10 post-challenge, one animal in group F died, and three control animals were euthanized after reaching the human endpoint value set at 13.

Conversely, from day 5 to day 10, only one vaccinated animal (group C) showed severe symptoms, which slowly disappeared, with full recovery.

Necroscopy performed on all control animals revealed spleen hyperplasia, enlargement of prescapular lymph nodes, lesions of the nasal mucosa, and scrotal nodules.

After the challenge test, the immune response against LSDV was analyzed in the vaccinated and control animals using ELISA and SN. In the vaccinated animal group C, the antibody titers started to increase from day 7 in ELISA and from day 3 in SN (Figure 9 and Figure 10), respectively, while in the control group F, the immune response was first detected at day 24 in ELISA (Figure 9), when one out the two survived control animals became positive, and at day 9 in SN (Figure 10).

The antibody response (IgM and IgG, Figure 11 and Figure 12) against KLH during the challenge was not significantly different from the antibody response before the challenge.

In the meantime, responses against KLH observed in the unchallenged group A (Figure 5A and Figure 6A) were similar to those in the challenged groups.

Regarding the γ-interferon, its concentration was significantly higher in vaccinated animals than in the control ones until day 9 (Figure 13). The peak observed in the control group on day 7 was an outlier, likely due to technical issues during the analytical step.

Next, when on day 10, one animal died, and the other three were euthanized, the statistical analysis was not applicable until the end of the challenge test, because of the lack of a sufficient number of animals.

Blood samples of all control animals tested positive for LSDV in PCR at least once, while only two vaccinated animals were found positive at PCR only on days 7–8 and days 9–14 (Figure 14A,B). The differences observed between the two groups were statistically significant (*p* < 0.05).

Within ten days post-challenge, the PCR analysis of ocular, nasal, and oral swabs yielded similar results in both groups, with multiple positive samples. The positivity rate was slightly higher in the control group than in the vaccinated one, in particular in nasal swabs, although this difference was not statistically significant (Figure 14).

## 4. Discussion

Inactivated vaccines, due to their safe profile, represent a preferred choice for fighting diseases in LSD-free countries. Furthermore, vaccination is the most useful tool against lumpy skin disease spread in European countries. So far, only live attenuated vaccines have been commercially available. Even if, compared to these last, inactivated vaccines seem to be less immunogenic, they are still being studied because of their safety, and promising results on their immunogenicity against Capripoxviruses have been published [15,39].

Our work aimed to develop and evaluate four different formulations of inactivated vaccines against lumpy skin disease. The vaccines produced were tested for safety and immunogenicity in four calf groups over 385 days, except for one formulation, whose efficacy was evaluated 68 days post-vaccination. The formulations produced differed from each other in terms of the cells used (continuous cells or primary cells) and the inclusion or not of an exogenous protein (KLH). Specifically, KLH was used as a positive marker to identify the vaccinated animals through an ad hoc ELISA.

After administrating the vaccines and placebos, some spikes of fever were recorded in a few animals belonging to all groups. Our results are similar to safety studies conducted by other researchers who tested inactivated vaccines against LSD. In particular, temperatures higher than 39.5 °C post-vaccination were found by several authors such as Haegeman [15], Hamdi [19], and Wolff [17]. In our study, the presence of spikes also in the control groups suggests that fever should be not directly attributable to the presence of viral antigen in the vaccines. Indeed, statistical analysis did not reveal any differences between the temperatures of the vaccinated and control groups E and F, which were inoculated with the placebo with and without KLH. To summarize, fever may be linked to the stress of handling animals during sampling or to the adjuvant used, which was also present in the two placebos. However, the safety profile of Montanide Gel reported by Matsiela [16] and also documented in our previous studies [40] suggests excluding Montanide Gel as a cause of fever and considering the constriction during sampling as a likely cause of fever spikes. In addition, our vaccines do not produce any local reaction, in contrast to what has been reported by some authors; for instance, the authors of [15,19] recorded reactions probably linked to the adjuvant used, which, in the case of Hamdi, was a water-in-oil emulsion. As in previous works, our vaccines confirm the knowledge regarding the safety profile of inactivated vaccines that, in the case of LSD, is a very important characteristic considering the adverse reactions resulting from the use of live attenuated vaccines against LSD.

The immunogenicity studies we conducted show the ability of our vaccines to induce both humoral and cell-mediated immune responses, which were recorded until day 385. Moreover, our results show the need to administer a booster dose to elicit an immune response detectable by SN. Our data seem to be similar to the results of Matsiela [16] regarding an inactivated vaccine tested on rabbits, containing an antigen dose equal to 10^5^ TCID_50_/mL, and to the data of Wolff [17] regarding two vaccines with diluted antigen (10^5^ and 10^4^ TCID_50_/mL before inactivation). By contrast, our data differ from those obtained in other previous studies [19,41], where the percentage of SN-positive animals nevertheless increased with the first vaccination. These different performances could hypothetically be due to, on the one hand, the different production methods used by researchers, and on the other, the type of adjuvants used in the vaccines. Regarding vaccine titers, our data show that differences less than 1 log in antigen concentration, as in the case of our vaccines produced in MDBK cells with respect to the vaccine produced in primary cells, are not involved in statistically significant differences in antibody response, according to what was previously found by Wolff [17].

In addition, concurrently with the humoral response, our vaccines seem to stimulate a rapid activation of cell-mediated immunity in the vaccinated groups. These results confirm what has been found not only by Hamdi [19], who observed a cellular-mediated response in an inactivated vaccine against LSDV, but also the results of some previous studies performed on vaccines against other viruses, such as porcine circovirus and those causing bovine respiratory disease [42,43], suggesting an effective cell-mediated immune response, which is usually reported for live attenuated vaccines. Our results are partially contrary to what was found by Haegeman [15], who revealed that the inactivated LSD vaccine elicited a transient cellular-mediated immune response in only 67% of the vaccinated animals.

Moreover, along with the immunogenicity elicited by the viral antigen, we demonstrated that the positive marker KLH allowed us to distinguish the vaccinated animals. As previously described, the antibody response against KLH increased early (day 7), and immunity against KLH was recorded until the end of the experiment. Although the KLH concentration in the MDBKv-KLH, PTLv-KLH, and placebo–KLH vaccines was equal, lower values of O.D. IgM and, in particular, IgG were found in some cases. The latter findings could likely be ascribed to the competition in the vaccines between viral proteins and KLH in stimulating the antibody response. Interestingly, during the challenge, the vaccinated animals remained identifiable through the ELISA for KLH by both IgM and IgG. These data meet the requirement of including a positive marker in a vaccine, as also documented by Ray [25], for the application of the DIVA strategy. In contrast with Ray, who documented an adjuvant property of KLH, in our study, the addition of such protein to two vaccine formulations did not enhance the strength of the antibody response or the cellular immune response against the viral antigen, when compared with the formulations without KLH. The absence of adjuvant effects from KLH in our vaccines could be linked to the presence of whole viruses and not haptens or small recombinant proteins, which can take advantage of the carrier activities of KLH.

The efficacy study proved the incapability of the formulation administered to group C in reducing the shedding of the virus, as evidenced by the positivity of ocular, nasal, and oral swabs in PCR results. In the meantime, the efficacy study demonstrated the properties of the vaccine produced and tested to reduce the clinical signs and viremia, as confirmed by the statistical analysis performed (*p* < 0.05). In this case, the early rise in the antibody response of vaccinated animals after infection (day 3 in SN and day 7 in ELISA) plays a crucial role in viremia reduction, unlike what occurred in the infected control animals, in which the first positive animals in SN were found on days 9 and 24 in ELISA, respectively. The delay of ELISA to detect positivity in infected animals with respect to the SN test is also described by the producer in the validation report. That finding could pose some doubt in the application of this test as a method for monitoring the disease in LSD-free countries. In addition, differences were also observed in the PCR results of ocular, nasal, and oral swabs. However, these findings were not statistically significant (*p* > 0.05), probably because of the small size of the groups. In particular, the percentage of positivity in the ocular, nasal, and oral swabs was higher in the control animals than in the vaccinated ones. Furthermore, the CT % values recorded in ocular, nasal, and oral swabs were lower in the control animals than in the vaccinated ones, indicating a reduction in virus replication in the vaccinated group after the challenge. The data regarding virus dissemination in the vaccinated animals after challenge confirm what was reported by Wolff [41] for one of the two inactivated vaccines adjuvanted with adjuvant B, where blood samples were found positive in PCR; however, our findings are in contrast with those of some authors such as Hamdi [19], who found only DNA traces on the skin biopsies of vaccinated animals, and Wolff [17,41] who developed some vaccines inducing complete sterilizing immunity. These differences could be due to the different production procedures used, as well as the challenge models employed. In our challenge protocol, the total amount of virus inoculated intravenously (10^7.3^ TCID_50_/mL) diluted in 5 mL was higher than the viral quantities used by Wolff for challenges (10^7.0^ TCID_50_ and 10^6^/mL, 10^6.9^ TCID_50_/mL). A unique validated model for the challenge and a unique strain to be used in all vaccine efficacy studies could help researchers in the development of safer and more efficacious vaccine formulations than ever. Consequently, further studies will be carried out in order to analyze the genome of the virus strain used and present the data obtained.

As a future direction, it could be interesting to increase the sample size in order to optimize the precision of some observed effects and underline some of the subtle differences found in our study, which were not statistically significant. In fact, the number of animals selected for this study allows us to demonstrate differences of only 80% between the groups under comparison.

## 5. Conclusions

To summarize, our work contributes to existing research by demonstrating the feasibility of a safe, immunogenic, and effective inactivated vaccine to prevent the spread of LSD in disease-free countries, avoiding the negative consequences related to live attenuated ones. Furthermore, we successfully applied for the first time a positive marker-based DIVA for LSD vaccine production, in order to fulfill one of the requirements set for a disease-free European country. Additional studies in lactating cows will provide further insights into the LSD vaccine safety contributing to the use of inactivated vaccines against LSD over attenuated ones.

## Figures and Tables

**Figure 1 vaccines-12-01302-f001:**
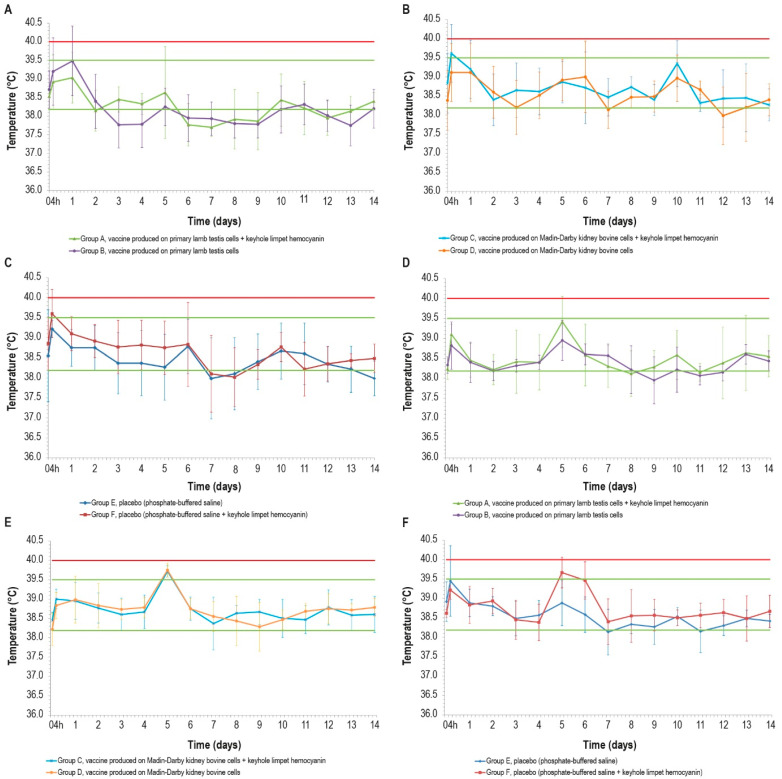
Mean rectal temperatures over time in the six cattle groups (6 animals/group) after the first and second doses: (**A**,**D**) vaccines produced in PLT cells with (group A) or without (group B) KLH; (**B**,**E**) vaccines produced in MDBK cells with (group C) or without (group D) KLH; (**C**,**F**) placebo with (group E) or without (group F) KLH; d_12_ data for one animal in group F are missing. Error bars: standard deviation. Red line: pyrexia. Green lines: range of physiological temperature.

**Figure 2 vaccines-12-01302-f002:**
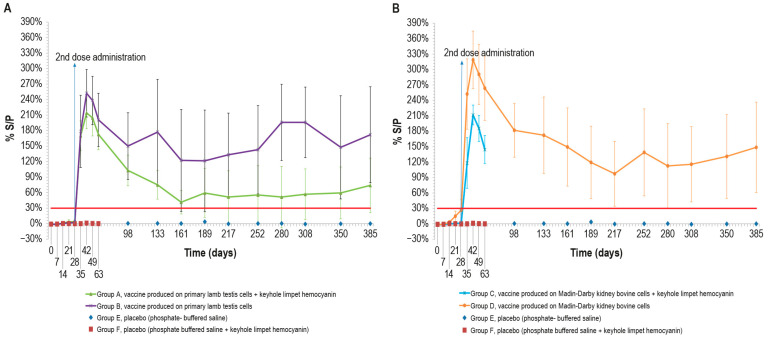
Mean LSDV ELISA titers (S/P %) measured over time, starting from the first vaccine inoculation (day 0) until the end of the study (day 385): (**A**) mean ELISA titers, expressed as S/P%), in animals inoculated with vaccine produced on PLT cells with (group A) and without KLH (group B); (**B**) mean ELISA titers in animals inoculated with vaccine produced on MDBK cells with (group C) and without KLH (group D). Both panels show the two control groups (E and F). The mean values of group C and group F (placebo) were recorded until day 63 when they underwent the challenge. Error bars: standard deviation; red line: cut-off.

**Figure 3 vaccines-12-01302-f003:**
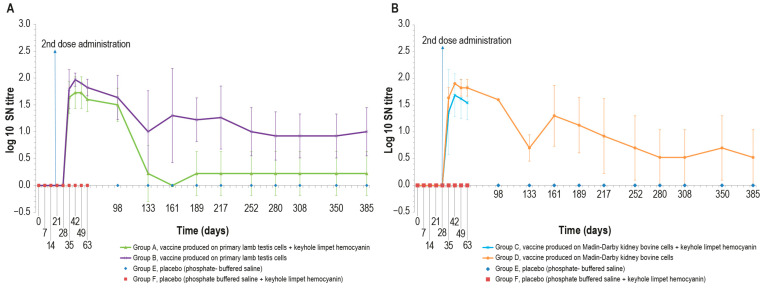
Mean serum LSDV neutralizing titers (log_10_) over time: (**A**) mean titers of animals inoculated with vaccine produced on PLT cells with (group A) or without (group B) KLH; (**B**) mean titers of animals inoculated with vaccine produced on MDBK cells with (group C) or without (group D) KLH. The two control groups (groups E-F) are included in both panels. The data concern the entire observational period (day 385) except for groups C and F, which underwent the challenge on day 63. The error bars indicate the standard deviation at each time point.

**Figure 4 vaccines-12-01302-f004:**
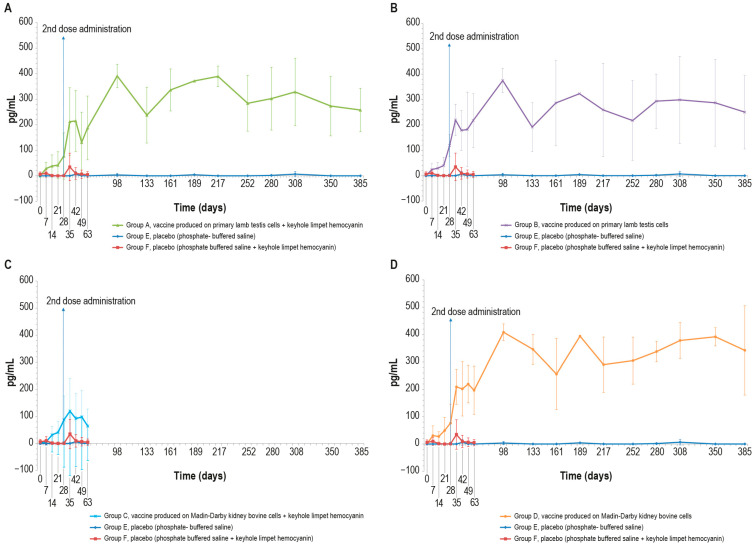
Mean interferon-γ values (pg/mL) measured in the six animal groups over time: (**A**) γ-interferon values in animals inoculated with vaccine produced on PLT cells with KLH (group A); (**B**) interferon-γ values in animals inoculated with vaccine produced on PLT cells without KLH (group B); (**C**) interferon-γ values of animals inoculated with vaccine produced on MDBK cells with KLH (group C); (**D**) interferon-γ values in animals inoculated with vaccine produced on MDBK cells without KLH (group D). The data concern the entire observational period (day 385), starting from the first vaccine inoculation (day 0), except for groups C and F that underwent the challenge at day 63. The two placebo groups (E and F) are included in all the panels. The error bars indicate the standard deviation at each time point.

**Figure 5 vaccines-12-01302-f005:**
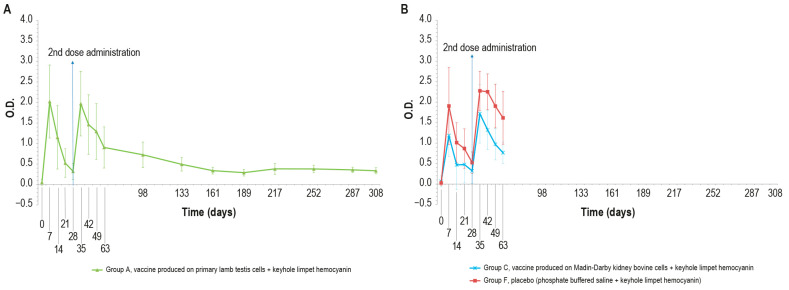
KLH IgM antibody titers, expressed as optical density absorbance at 600 nm, measured over time in the animal groups inoculated with formulations containing the exogenous protein KLH at day 0 and day 28: (**A**) IgM titers in animal group inoculated with vaccine produced on PLT cells with KLH (group A); (**B**) IgM titers in animal groups inoculated with vaccine produced on MDBK cells with KLH (group C) and PBS with KLH (group F). The error bars indicate the standard deviation. Groups C and F underwent the challenge on day 63.

**Figure 6 vaccines-12-01302-f006:**
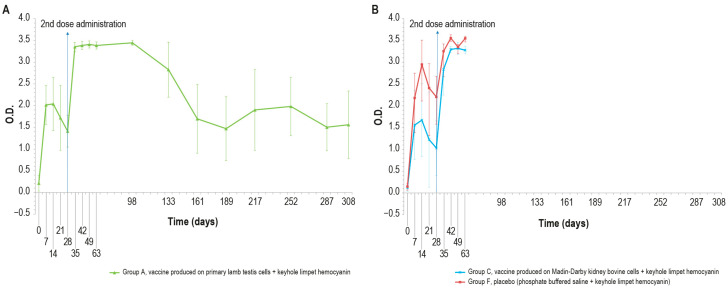
IgG KLH antibody titers, expressed as optical density absorbance at 600 nm, measured over time in the animal groups inoculated with formulations containing the exogenous protein KLH at day 0 and day 28: (**A**) IgG titer in the animal group inoculated with vaccine produced on primary lamb testis cells with KLH (group A); (**B**) IgG titers in animal groups inoculated with vaccine produced on MDBK cells with KLH (group C) and PBS with KLH (group F). The error bars indicate the standard deviation. Groups C and D underwent the challenge on day 63.

**Figure 7 vaccines-12-01302-f007:**
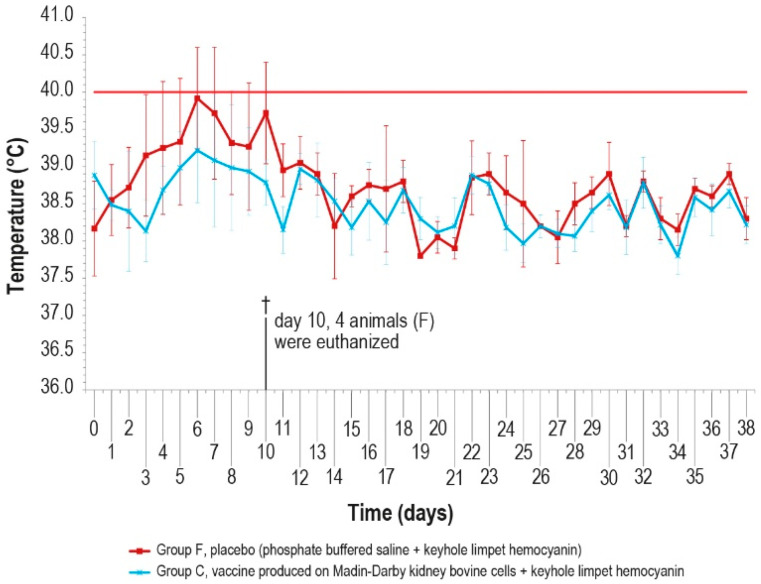
Mean rectal temperatures measured, after challenge (day 0), in group C, inoculated with vaccine produced in MDBK cells with KLH, and group F inoculated with PBS and KLH (placebo). Error bars indicate the standard deviation. Temperatures above the horizontal red line were considered pyrexia. Four animals belonging to group F were euthanized (†) on day 10.

**Figure 8 vaccines-12-01302-f008:**
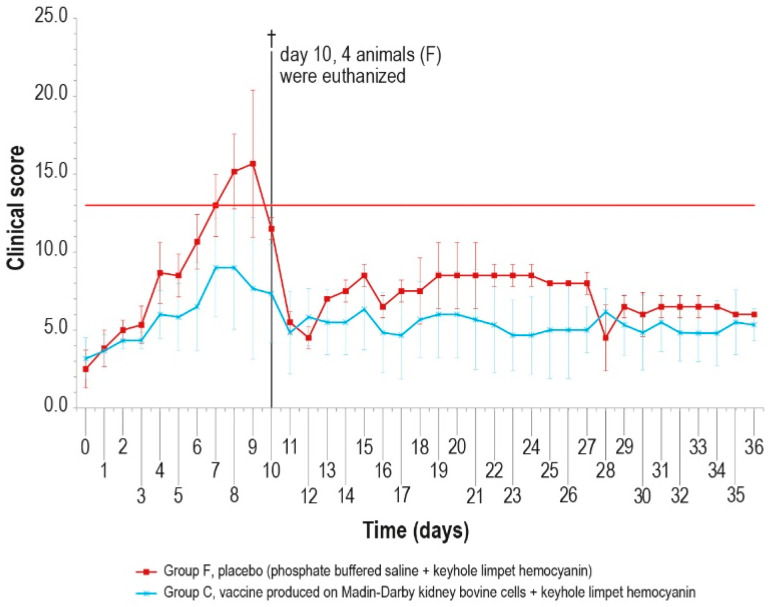
Mean clinical score values after challenge in the two animal groups (group C and group F). On day 10, four animals were euthanized (†). The error bars represent the standard deviation. The red line represents the human endpoint value set to 13.

**Figure 9 vaccines-12-01302-f009:**
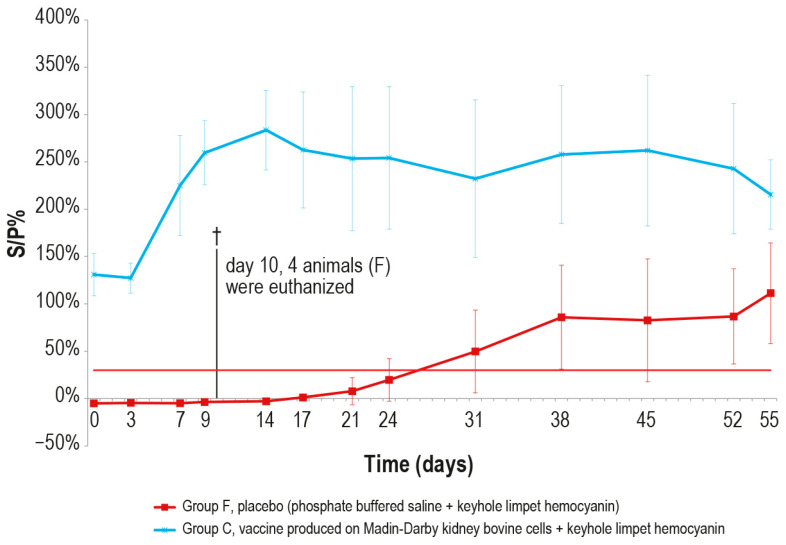
Mean LSDV ELISA titers, expressed as S/P %, measured after the challenge (day 0) in groups C and F (vaccinated and placebo, respectively). The cut-off of the test was set at 30% (red line). On day 10, 4 animals belonging to the control group F were euthanized (†).

**Figure 10 vaccines-12-01302-f010:**
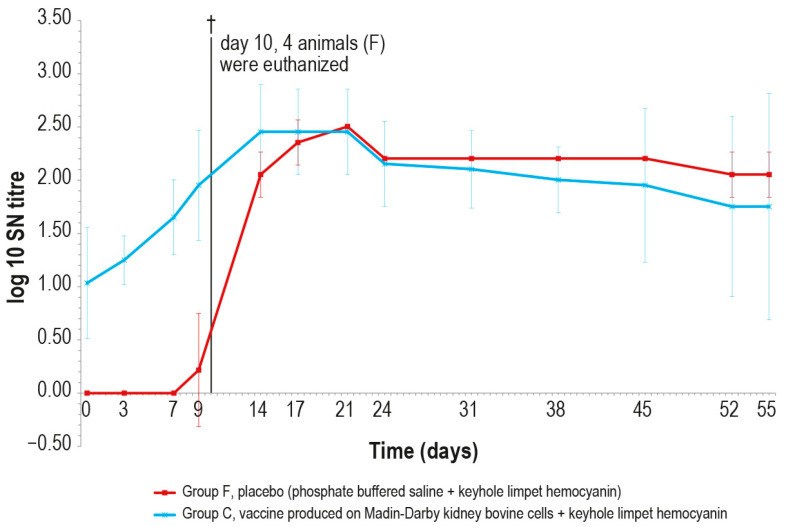
Mean serum LSDV neutralizing titers, expressed as log_10_ of the reciprocal of the last positive dilution tested, measured after challenge (day 0), in groups C and F (vaccinated and placebo, respectively). The error bars indicate the standard deviation at each time. Four animals belonging to group F were euthanized (†) on day 10.

**Figure 11 vaccines-12-01302-f011:**
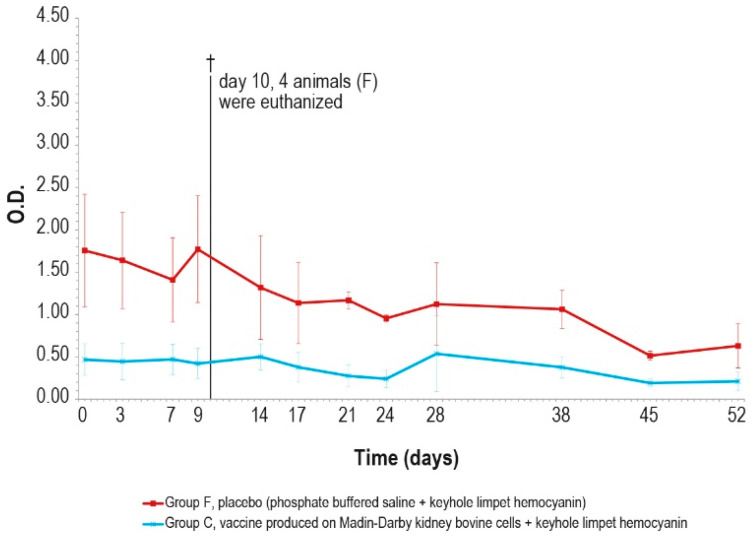
KLH IgM antibody titers, expressed as optical density absorbance at 600 nm, measured after challenge (day 0) in groups C and F (vaccinated and placebo, respectively). The error bars indicate the standard deviation. On day 10, 4 animals belonging to the control group F were euthanized (†).

**Figure 12 vaccines-12-01302-f012:**
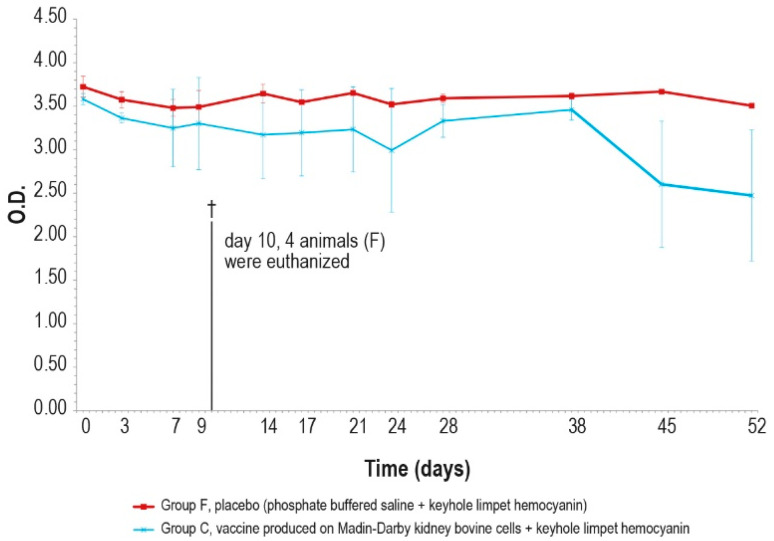
KLH IgG antibody titers, expressed as optical density absorbance at 600 nm, measured after challenge (day 0) in groups C and F (vaccinated and placebo, respectively). The error bars indicate the standard deviation. On day 10, 4 animals belonging to the control group F were euthanized (†).

**Figure 13 vaccines-12-01302-f013:**
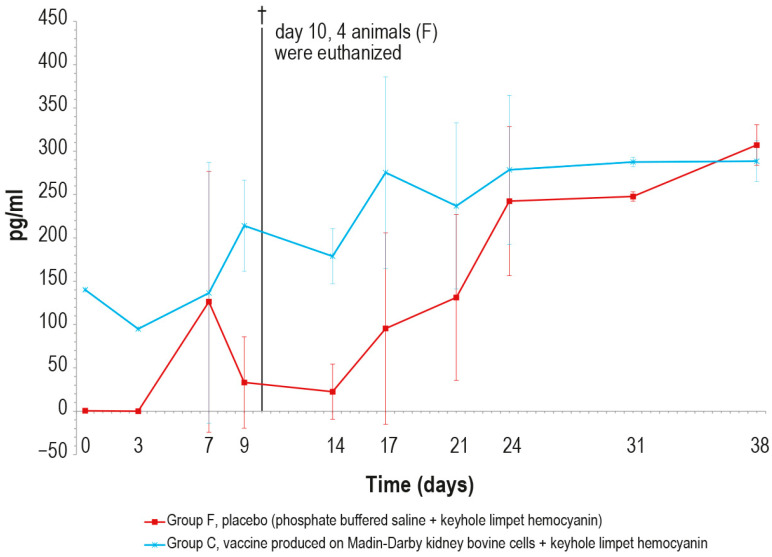
Mean γ-interferon values (pg/mL) measured after the challenge in the animal groups C and F (vaccinated and placebo, respectively). The error bars indicate the standard deviation at each time. Four animals belonging to group F were euthanized (†) on day 10.

**Figure 14 vaccines-12-01302-f014:**
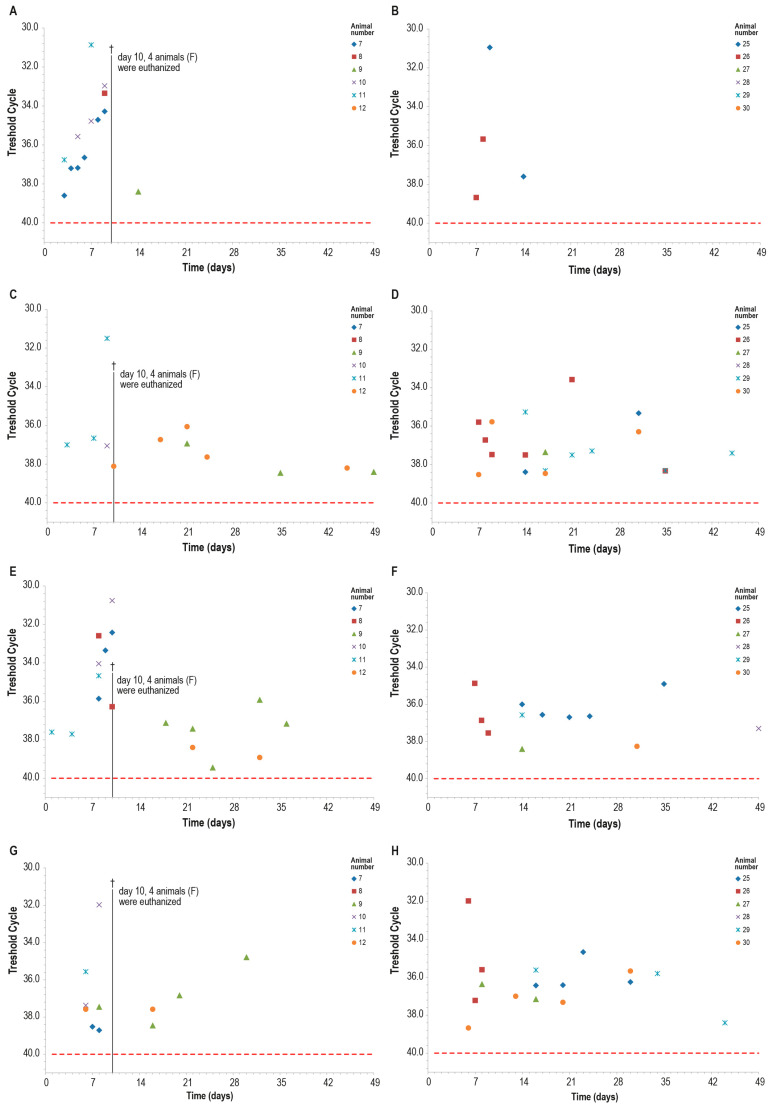
PCR for capripox viruses in different samples in the two animal groups (group C—panels (**B**,**D**,**F**,**H**) and group F—panels (**A**,**C**,**E**,**G**)): (**A**,**B**) blood PCR; (**C**,**D**) ocular swabs PCR; (**E**,**F**) nasal swabs; (**G,H**) oral swabs. On day 10, four animals in the control group F were euthanized (†). The red line indicates the threshold cut-off, which was set to 40.

**Table 1 vaccines-12-01302-t001:** Formulations administered to the six animal groups. The formulations differed from each other in terms of the cell line used for virus amplification and the presence or not of KLH. The symbols + and − indicate the presence or not of the substance in the column headers.

Formulation	Name	Animal Group	PBS	Cell Line for Virus Replication		Antigen		Montanide Gel	Saponin
				PLT	MDBK	LSDV	KLH		
I	PLTv-KLH	A	−	+	−	+	+	+	+
II	PLTv	B	−	+	−	+	−	+	+
III	MDBKv-KLH	C	−	−	+	+	+	+	+
IV	MDBKv	D	−	−	+	+	−	+	+
V	placebo	E	+	−	−	−	−	+	+
VI	Placebo–KLH	F	+	−	−	−	+	+	+

## Data Availability

All data acquired are shown in the paper.

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
