# Peer review of "The Safety and Efficacy of New DIVA Inactivated Vaccines Against Lumpy Skin Disease in Calves"

_vaccines, 2024, doi:10.3390/vaccines12121302_

Round 1

Reviewer 1 Report

Comments and Suggestions for Authors

The manuscript by Ronchi et al. discusses the development and evaluation of a new DIVA inactivated vaccines against lumpy skin disease in calves. The research utilized keyhole limpet hemocyanin (KLH) as a marker to differentiate vaccinated from infected cattle. Four distinct inactivated vaccines were produced and one was further tested through a challenge trial. Although the inactivated vaccine showed good results in serological tests, below are some comments and concerns regarding the study.

1. There is great difficulty in seeing all figures clearly. Make the font size larger.

2. The representation of physiological temperature in Figure 1 should be marked with a normal range not just a green line.

3. The date of Figure 4c is not clear and misleading, and even if this indicator is not measured after challenge, the time should be unified with the other three groups when presenting the results. Other pictures have similar problems.

4. Could the authors explain in KLH antibodies analysis, why the O.D. values in control group F were higher than in the vaccinated group? They have the same KLH concentration which equal to 130 µg/mL.

5. The final vaccine was MDBKv-KLH, but there was no explanation why PLTv-KLH was not used and why challenge tests were not conducted.

6. Although the animals in group C did not have fever and had good clinical scores, the PCR analysis of blood and ocular, nasal, and oral swabs were able to detect the virus shedding, which does not seem to indicate that the vaccine provides sufficient protection.

7. More data or literature is needed to support the conclusion that this vaccine can stimulate a rapid activation of the cell-mediated immunity in vaccinated groups. Hamdi’s study alone is not enough to prove it.

8. If possible, add some histopathological results to enrich the experimental results.

9. There are small errors on lines 400-401, 436-439, 476, 486-489, and in the title of Figure 13.

Author Response

For research article

Safety and efficacy of a new DIVA inactivated vaccines against lumpy skin disease in calves

Response to Reviewer 1 Comments

1. Summary

Dear reviewer,

Thank you so much for all your suggestions which really help to make all the work we have done clearer. We tried to modify all the points you highlighted with the exception of some modifications that need to plan new activities such as histopathological results. We are working on a second article regarding the same vaccine tested on lactating cows where we hope to integrate the new data that we will acquire in the next activities.

We modified all the figures accordingly with your suggestions in order to clarify the results section.

Best Regards

Gaetano Federico Ronchi & Mariangela Iorio

2. Questions for General Evaluation

Reviewer’s Evaluation

Response and Revisions

Does the introduction provide sufficient background and include all relevant references?

Can be improved

We modified the introduction adding new bibliography regarding inactivated LSD vaccines.

Are all the cited references relevant to the research?

We add references

Is the research design appropriate?

Can be improved

We highlighted the limit of the study (624-628)

Are the methods adequately described?

Yes

Are the results clearly presented?

Must be improved

All the figures were modified in order to clearly present the data

Are the conclusions supported by the results?

Can be improved

We modified the discussion section

3. Point-by-point response to Comments and Suggestions for Authors

  1. Comments 1: There is great difficulty in seeing all figures clearly.Make the font size larger

Response 1: All figures were modified accordingly

  1. Comments 2: The representation of physiological temperature in Figure 1 should be marked with a normal range not just a green line.

Response 2: The two panels of figure 1 were modified accordingly; in material and method we modified the description (lines 189-191) and we did the same in the caption of figure 1.

  1. Comments 3: The date of Figure 4c is not clear and misleading, and even if this indicator is not measured after challenge, the time should be unified with the other three groups when presenting the results. Other pictures have similar problems.

Response 3: All the dates were unified in all figures

  1. Comments 4: Could the authors explain in KLH antibodies analysis, why the O.D. values in control group F were higher than in the vaccinated group? They have the same KLH concentration which equal to 130 µg/mL.

Response 4: We added a hypothetical explanation in the discussion paragraph. (lines 579-583)

  1. Comment 5: The final vaccine was MDBKv-KLH, but there was no explanation why PLTv-KLH was not used and why challenge tests were not conducted.

Response 5: We added an explanation in the results paragraph 3.4.1 Efficacy study (lines 437-444)

  1. Comment 6: Although the animals in group C did not have fever and had good clinical scores, the PCR analysis of blood and ocular, nasal, and oral swabs were able to detect the virus shedding, which does not seem to indicate that the vaccine provides sufficient protection.

Response 6: In the discussion we added consideration regarding virus shedding and the incapability of our vaccine in prevent it (lines 593-597)

  1. Comment 7: More data or literature is needed to support the conclusion that this vaccine can stimulate a rapid activation of the cell-mediated immunity in vaccinated groups. Hamdi’s study alone is not enough to prove it.

Response 7: We added some references about cell-mediated immune response in inactivated vaccines and we add the verb “seem” in order to make less strong the conclusion that the vaccine can stimulate the rapid cell mediated immune response (lines 568-573)

  1. Comment 8: if possible, add some histopathological results to enrich the experimental results.

Response 8: Unfortunately we do not have the possibility to perform a histopathological study.

  1. Comment 9: There are small errors on lines 400-401, 436-439, 476, 486-489, and in the title of Figure 13

Response 9: We corrected as suggested

4. Response to Comments on the Quality of English Language

Point 1:

Response 1:    /

5. Additional clarifications

/

For review article

Response to Reviewer X Comments

1. Summary

Thank you very much for taking the time to review this manuscript. Please find the detailed responses below and the corresponding revisions/corrections highlighted/in track changes in the re-submitted files. [This is only a recommended summary. Please feel free to adjust it. We do suggest maintaining a neutral tone and thanking the reviewers for their contribution although the comments may be negative or off-target. If you disagree with the reviewer's comments please include any concerns you may have in the letter to the Academic Editor.]

2. Questions for General Evaluation

Reviewer’s Evaluation

Response and Revisions

Is the work a significant contribution to the field?

[Please give your response if necessary. Or you can also give your corresponding response in the point-by-point response letter. The same as below]

Is the work well organized and comprehensively described?

Is the work scientifically sound and not misleading?

Are there appropriate and adequate references to related and previous work?  

Is the English used correct and readable?  

3. Point-by-point response to Comments and Suggestions for Authors

Comments 1: [Paste the full reviewer comment here.]

Response 1: [Type your response here and mark your revisions in red] Thank you for pointing this out. I/We agree with this comment. Therefore, I/we have.[Explain what change you have made. Mention exactly where in the revised manuscript this change can be found – page number, paragraph, and line.]

“[updated text in the manuscript if necessary]”

Comments 2: [Paste the full reviewer comment here.]

Response 2: Agree. I/We have, accordingly, done/revised/changed/modified…..to emphasize this point. Discuss the changes made, providing the necessary explanation/clarification. Mention exactly where in the revised manuscript this change can be found – page number, paragraph, and line.]

“[updated text in the manuscript if necessary]”

4. Response to Comments on the Quality of English Language

Point 1:

Response 1:    (in red)

5. Additional clarifications

[Here, mention any other clarifications you would like to provide to the journal editor/reviewer.]

Reviewer 2 Report

Comments and Suggestions for Authors

The manuscript by Ronchi et al., provides valuable insights into the development of a DIVA inactivated vaccine for LSDV, with promising results in safety and immunogenicity. Addressing the critical points below will enhance clarity, strengthen the scientific rationale, and improve the manuscript's overall impact.

Comment #1) The abstract should accurately represent the study's full scope, but there are areas that need improvement in clarity. For instance, the description of the challenge trial and its outcomes is vague. The authors mention that four out of six control animals were euthanized, but there is no information on the vaccinated groups. Specifically, did the vaccines prevent clinical signs or reduce the severity of the disease in a significant way? Including these results will strengthen the abstract. Please revise accordingly.

Comment #2) The choice of PLT and MDBK cell lines lacks an explicit justification. It would strengthen the study to include why both were chosen and if there were advantages in viral yield, immune response stimulation, or other factors. Similarly, a rationale for using only male calves should be discussed, considering immune responses could vary with sex​.

Comment #3) The manuscript references the use of a field virus strain isolated in 2017; however, complete identification and characterization of the isolate are not presented. Detailing the strain's properties would add robustness to the study by ensuring the isolate’s relevance and comparability to circulating strains​.

Comment #4) In Table 1, formulation E (placebo without KLH) is said to contain only PBS. This contradicts later statements linking temperature spikes to Montanide adjuvant. This section could be clearer by specifying all placebo components to avoid confusion.​

Comment #5) Figures 1 to 5 are of poor resolution, making it challenging to discern finer details. Revisions for high-quality images with clear legends and labels would enhance readability​.

Comment #6) The way the γ-interferon values are currently presented is not informative enough. Displaying all groups in a single panel or a clearer format would improve data interpretation and allow easier comparison across groups​.

Comment #7) As a limitation, the study's sample size is acknowledged to be small, potentially affecting statistical power and the precision of observed effects. This limitation should be noted in the manuscript as it impacts the reliability of the conclusions.

Minor comments

Line 12: add word (disease) to the name of the virus, It is Lumpy Skin Disease Virus (LSDV).

Line 12 and 38: The family, genus and species names should be italicized throughout the manuscript to adhere to standard scientific notation​.

Line 18: Replace the word (amplified) with (propagated or cultured) because it is more scientifically precise.

Round 2

Reviewer 2 Report

Comments and Suggestions for Authors

The authors have done a good job addressing all concerns and comments.